# Real-World Case Series of Efgartigimod for Japanese Generalized Myasthenia Gravis: Well-Tailored Treatment Cycle Intervals Contribute to Sustained Symptom Control

**DOI:** 10.3390/biomedicines12061214

**Published:** 2024-05-30

**Authors:** Shingo Konno, Takafumi Uchi, Hideo Kihara, Hideki Sugimoto

**Affiliations:** Department of Neurology, Toho University Ohashi Medical Center, Tokyo 153-8515, Japan; m09014u@gmail.com (T.U.); hideo.kihara@med.toho-u.ac.jp (H.K.); sugi-h@oha.toho-u.ac.jp (H.S.)

**Keywords:** myasthenia gravis, efgartigimod, individualized treatment, real-world, cycle interval

## Abstract

Introduction: Myasthenia gravis (MG), an immune disorder affecting nerve-muscle transmission, often necessitates tailored therapies to alleviate longitudinal symptom fluctuations. Here, we aimed to examine and compare the treatment cycle intervals and efficacy of efgartigimod in four patients. This case series mainly offers insights into personalized treatment cycle intervals and the efficacy of efgartigimod for patients with MG in our facility in Japan. Methods: We retrospectively analyzed four patients with MG (2 patients with early-onset, 1 with late-onset, and 1 with seronegative MG, mainly managed with oral immunosuppressants as prior treatments) who completed four or more cycles of efgartigimod treatment from January 2022 to September 2023. We focused on changes in serum immunoglobulin (IgG) level, acetylcholine receptor antibody (AChR-Ab) titer, and quantitative MG (QMG) score. Results: Efgartigimod, administered at a median of 5.0 [IQR 5.0, 7.5] weeks between cycles, led to decreased serum IgG levels in all patients and reduced AChR-Ab titers in seropositive patients. All patients showed sustained MG symptom improvement, with considerably reduced QMG scores before efgartigimod treatment. None of the patients required rescue medications or developed treatment-related adverse events. Conclusions: Customized efgartigimod administration intervals effectively enhanced clinical outcomes in patients with MG without notable symptom fluctuations, demonstrating the benefits of individualized treatment approaches and validating the safety of efgartigimod during the study period.

## 1. Introduction

Myasthenia gravis (MG) is an autoimmune disorder that affects the neuromuscular transmission, leading to muscle weakness and fatigability. It is caused by autoantibodies that target proteins, mainly acetylcholine receptor (AChR) and muscle-specific receptor tyrosine kinase (MuSK) on the postsynaptic muscle membrane. As a result, patients with MG often experience their muscles getting tired and weak, particularly after prolonged use. Resting can help improve this muscle weakness. Common symptoms of MG include ptosis, double vision, weakness of the face, and difficulty in swallowing and controlling arm and leg movements. MG is related to the immune system, mostly involving B cells and the generation of particular antibodies, such as AChR antibody (AChR-Ab) and MuSK antibody (MuSK-Ab) [1]. The diversity of MG is further delineated by its classification into subtypes based on onset age, antibody status, and thymic pathology, each presenting unique management challenges. In terms of the subtypes we present, generalized early-onset MG (g-EOMG) typically manifests before age 50 and is often associated with thymic hyperplasia, with patients commonly having antibodies against the acetylcholine receptor. In contrast, generalized late-onset MG (g-LOMG) appears after the age of 50 and often lacks thymic abnormalities, potentially presenting more severe symptoms and differing responses to treatments compared to earlier-onset forms. Another subtype, generalized seronegative MG (g-SNMG), is marked by the absence of typical AChR and MuSK antibodies, complicating both diagnosis and treatment due to the lack of specific serological markers [2].

These subtypes highlight the clinical heterogeneity within MG, emphasizing the necessity for personalized treatment strategies to address the diverse manifestations of the disease effectively. Recent advancements in treatment have significantly influenced management strategies for MG. The 2020 update by the Myasthenia Gravis Foundation (MGFA) has been particularly impactful, emphasizing the need for tailored treatment [3]. This guideline update recommends specific treatments such as rituximab, eculizumab, and methotrexate and emphasizes the importance of early immunosuppression in cases of ocular MG. Reflecting major progress based on the latest clinical evidence, these guidelines aim to enhance patient outcomes through more customized and effective therapeutic regimens.

Some reviews have highlighted that patients with MG can be classified based on antibody status and clinical presentation, and treatment responses may vary accordingly [4,5]. These reviews also note the trend toward minimizing the use of high-dose corticosteroids for long durations [4,5]. The importance of thymectomy in improving clinical outcomes in patients with early-onset non-thymomatous MG with AChR-Ab has also been highlighted, along with the approval of new drugs such as eculizumab, efgartigimod, and ravulizumab by the U.S. Food and Drug Administration (FDA) for adult patients with generalized MG who are AChR-Ab positive [6].

Recent advances in MG management have led to a substantial decline in the mortality of patients. This improvement is attributed to evolving interventions in critical care and medical management. In the past few years, there have been changes in the standard of care for MG, including the approval of new medications for refractory MG, with several promising drugs in clinical trial stages [7].

In a comprehensive summary of current therapies for MG, corticosteroids reportedly improve prognosis and reduce mortality rate. However, for many patients suffering from steroid-related adverse effects, current treatment strategies recommend a successful combination of plasmapheresis, immunosuppressive therapy, thymectomy, and targeted immunomodulatory agents [8,9]. Efgartigimod, a targeted immunomodulatory agent, blocks the neonatal Fc receptor (FcRn). This receptor is responsible for the prolonged lifespan of IgG antibodies. By inhibiting FcRn, efgartigimod accelerates the degradation of IgG antibodies, including the pathogenic autoantibodies implicated in the development and progression of MG [10]. The U.S. FDA has approved efgartigimod for adult patients with generalized MG who are AChR-Ab positive; it has also been approved for patients with or without AChR-Ab, including MuSK-Ab, and who are seronegative in Japan [11]. The ADAPT trial, conducted between 2018 and 2019, emphasized the safety, efficacy, and tolerability of efgartigimod in patients with generalized MG. This study marked a pivotal point in the treatment of MG, showcasing the effectiveness of efgartigimod in considerably improving patient outcomes compared to placebo [12]. In the study, patients received efgartigimod in treatment cycles of 4 once-weekly intravenous infusions (10 mg/kg) per cycle, with a median of 7.3 weeks between cycles (the period from the last infusion in the previous cycle to the first infusion in the subsequent cycle) [12,13]. Whereas, in the ADAPT+ study, the mean cycle duration was 61.4 days (40.4 days between cycles) [14,15]. Both the ADAPT study and ADAPT+ study, the open-label extension study, are characterized by individualizing treatment cycles based on clinical response. This approach allows the treatment interval to be tailored to each patient’s response. However, the ADAPT/ADAPT+ trial protocols required that the subsequent treatment cycle be allowed when the improved MG Activities of Daily Living (MGADL) scale returned to nearly baseline level and that treatment cycles be maintained consistently at intervals of no less than 4–5 weeks. This means symptom fluctuations have a profound effect on the lives of people with MG [16]. Minimizing symptom fluctuations in patients receiving efgartigimod for generalized MG is imperative for optimizing treatment efficacy and enhancing QOL. The proposed study seeks to fill existing knowledge gaps by evaluating the real-world impact of efgartigimod on symptom stability and patient outcomes. Insights gained could lead to improved treatment strategies and patient care practices, ultimately contributing to the broader understanding of managing autoimmune conditions.

In this study, we aimed to examine and compare the treatment cycle intervals and efficacy of efgartigimod in four patients against the known findings from the ADAPT and ADAPT + studies. This study will lead to the development of personalized administration strategies for efgartigimod in the clinical setting.

## 2. Case Description

Herein, we describe four cases of MG treated with efgartigimod. From January 2022 to September 2023, 97 patients with MG visited our hospital. Of these, six patients received efgartigimod treatment. Two patients who received less than two cycles of efgartigimod were excluded. Consequently, four patients who completed four or more cycles of efgartigimod treatment were included in the study (Figure 1).

### 2.1. Case 1

A 40-year-old woman diagnosed with generalized, early-onset MG, who had not undergone thymectomy, was managed with tacrolimus and prednisolone for a triennial period, commencing from the initial manifestation of symptoms to the initiation of efgartigimod therapy (Figure 2A). This therapeutic regimen engendered a notable improvement in her QMG score defined by MGFA [17]; it declined from an initial value of 15 to 9 points. Subsequently, she received five cycles of efgartigimod at 10 mg/kg, administered at an inter-cycle interval of 5.0 [4.0, 10.5] weeks (median [interquartile range (IQR [25%, 75%])]). Concurrently, there was a gradual diminution in her serum IgG level from a peak of 1264 to under 809 (the lowest level of 547) mg/dL, AChR-Ab level from 76 to 32 nmol/L, and QMG score from 9 to 5 points. Throughout the observational period, no alterations were made to her existing oral medication regimen, and there were no adverse events associated with efgartigimod. Similarly, no exacerbations occurred during the inter-cycle intervals. Owing to the alleviation of symptoms, the administration interval was extended to 15 weeks, during which the QMG score increased from 5 to 8 points. Re-starting efgartigimod improved her symptoms to minimal manifestation.

### 2.2. Case 2

A 78-year-old woman with a diagnosis of systemic seronegative MG and a history of thymectomy was treated with prednisolone for 2 years from the time of diagnosis, followed by tacrolimus for 15 years, and finally, efgartigimod was introduced at 10 mg/kg. This treatment strategy resulted in a substantial improvement in the QMG score, dropping from 8 to 6 points (Figure 2B). Efgartigimod was administered for five cycles with a median administration interval of 5.0 [5.0, 5.0] weeks. At the same time, a marked decrease in IgG level was observed, plummeting from a peak of 1924 mg/dL in the weeks before efgartigimod treatment to 802 (the lowest level of 723) mg/dL. The drop in the QMG score was only by 2 points. However, the patient suffered from droopy eyelids and cervical muscle weakness, which were considerably reduced after treatment with efgartigimod. During the observation period, the oral medication regime of the patient was not changed; no adverse effects attributable to efgartigimod were recorded; and no exacerbations occurred during the treatment interval.

### 2.3. Case 3

A 65-year-old man diagnosed with generalized late-onset MG, who had not undergone thymectomy, had received a 5-year cyclosporine regimen from the time of diagnosis, followed by a 3-year tacrolimus course until the initiation of efgartigimod therapy (Figure 2C). Despite immunosuppressant monotherapy interventions, the patient’s QMG score demonstrated substantial variability, ranging from 18 to 13 points. Efgartigimod at 10 mg/kg was dispensed at median intervals of 8.0 [6.0, 10.0] weeks for four cycles; however, the patient was found to have a gastric ulcer during the second injection of the third treatment cycle. Owing to the necessity of targeted ulcer treatment, the remaining two injections of this cycle were canceled, and the fourth treatment cycle was deferred by 10 weeks from the last administration. Serial measurements revealed a consistent decline in the IgG level from a peak of 1116 to a lowest of 440 mg/dL and AChR-Ab level from 640 to 232 nmol/L, and the lowest level was 160 nmol/L; the QMG score decreased from 13 to 8 points. The patient initially presented with dyspnea, which is a challenging breathing condition to manage, and marked fatigability, indicating a pronounced decrease in energy affecting the patient’s ability to carry out daily activities. Following the administration of efgartigimod, there was a gradual improvement in these symptoms, indicating a positive response to the treatment. However, the improvement was not steady or linear; instead, it was characterized by fluctuating patterns. During the observational period, no alterations were made to his oral medication regimen, and there were no adverse reactions attributable to efgartigimod. Thereafter, it was ascertained that the gastric ulcer was caused by *Helicobacter pylori* (*H. pylori*) infection.

### 2.4. Case 4

An 86-year-old man with generalized late-onset MG, who had not undergone thymectomy, was administered a tacrolimus regimen commencing at the time of diagnosis through to the initiation of efgartigimod therapy (Figure 2D). Throughout this period, the patient manifested pronounced dysphagia and consequential weight loss, necessitating two rounds of immunoadsorption and intravenous methylprednisolone administration at 1000 mg/day. Although these interventions elicited transient symptomatic relief, the patient’s condition deteriorated in association with elevated levels of AChR-Ab. Following the initiation of efgartigimod at 10 mg/kg, the QMG score was 11 points, concurrent with severe dysphagia, and the patient was subsequently subjected to six cycles of the medication, at a median treatment interval of 6.0 [5.0, 7.0] weeks. Notably, the IgG level showed a marked decline from a peak of 1200 to the lowest level of 537 mg/dL and the AChR-Ab level from 56 to 22 nmol/L; the QMG score reduced from 11 to 6 points. Notably, the patient did not require any immediate-acting treatments, such as plasmapheresis and intravenous methylprednisolone, and displayed a reversal in weight loss. Throughout the observational period, there were no modifications in his oral medication regimen nor were there any adverse reactions associated with efgartigimod administration.

## 3. Discussion

The commonalities among the four cases are a diagnosis of MG treated with a combination of immunosuppressive therapy and efgartigimod. Not all patients had undergone thymectomy. The patients were managed with oral immunosuppressants such as glucocorticoids, tacrolimus, and cyclosporine for varying durations before initiating efgartigimod therapy. The administration of efgartigimod resulted in an overall decline in the serum IgG level in all patients and AChR-Ab titer in seropositive patients. Furthermore, a significant improvement in MG symptoms was observed in all patients, as evidenced by a decrease in their QMG scores. During the observational periods, the oral medication regimens were unchanged. There were no adverse events specifically attributable to efgartigimod nor were there any exacerbations that needed other rescue treatment during the treatment cycle intervals of efgartigimod.

Efgartigimod distinguishes itself from other treatment options such as intravenous gamma globulin (IVIg), plasmapheresis, B-cell depletive therapies, and complement inhibitors due to its unique mechanism of action and clinical advantages. IVIg, despite its effectiveness, faces challenges like high production costs and dependence on volunteer blood donors for plasma extraction, leading to limited availability for treating various medical conditions like chronic inflammatory demyelinating polyneuropathy and hematological disorders. In contrast, efgartigimod rapidly reduces total IgG and pathogenic antibody levels in the serum, offering quicker and more immediate improvement compared to IVIg and other immunotherapies. Similarly to plasmapheresis, efgartigimod lowers IgG levels without the logistical issues associated with venous access, providing a smoother treatment experience. Unlike B-cell depletive therapies and severe immunosuppression, which carry the risk of increased infections and complications due to extensive immunosuppression, efgartigimod achieves IgG reduction without inducing such profound immunosuppression, potentially offering a safer treatment option. Additionally, while complement inhibitors are effective, they require extensive vaccination protocols due to their specific mechanism of action, whereas efgartigimod does not necessitate such vaccinations, making it a more convenient choice for patients and healthcare providers.

Overall, despite varying underlying conditions and ages of the patients, efgartigimod treatment improved the disease marker antibody levels and symptoms without requiring rescue treatment or adjustment of existing oral medications.

Sequential administration of efgartigimod decreases the IgG level [18]. Determining a safe range of IgG level is essential to avoid adverse effects of efgartigimod treatment. In the ADAPT+ study, a sustained decrease in the IgG level did not increase the prevalence of infections [14]. Furthermore, the ADVANCE study evaluated the efficacy and safety of the neonatal Fc receptor inhibitor efgartigimod in a treatment schedule of once per week or every other week for adults with primary immune thrombocytopenia [19]. In the ADAPT+ study, a repetitive decrease in IgG levels did not increase the prevalence of infections with subsequent cycles [14]. Notably, the ADVANCE study, which evaluated the efficacy and safety of the neonatal Fc receptor inhibitor efgartigimod in a treatment schedule of once per week or every other week for adults with primary immune thrombocytopenia [19], demonstrated that even with a sustained 60% decrease from baseline in total IgG, no increase in clinically important infections occurred as compared to placebo. Efgartigimod, which selectively targets IgG antibodies, likely spares other critical immune system elements such as innate and cellular immunity, thereby possibly reducing the risk of infections [20]. Nevertheless, patients with IgG levels less than 100 mg/dL for prolonged periods reportedly have an increased risk of recurrent and sometimes life-threatening infectious disease episodes [21]. The lowest IgG level in our patients was more than 100 mg/dL, but the patient of Case 3 developed a gastric ulcer caused by *H. pylori* infection. IgG antibodies contribute to the immune response against *H. pylori*; however, low IgG levels alone do not appear to definitively aggravate the infection based on the previous studies [22]. The role of other antibody isotypes like IgA seems important as well [23]. Therefore, we do not believe that low IgG associated with efgartigimod treatment can be ruled out as the cause of gastric ulcers. This selective mechanism of efgartigimod, which effectively reduces specific antibody levels without broadly suppressing the immune system, opens up new therapeutic possibilities. It mainly offers an alternative for patients who require immunotherapy but are unable or unwilling to tolerate the adverse effects commonly associated with corticosteroid therapy. This aspect of efgartigimod has also been underscored in a previous report [24].

When determining the appropriate treatment cycle intervals for efgartigimod in treating MG, patients’ needs must be considered. The treatment cycle schedule should ideally balance the therapeutic benefits of the drug in reducing the symptomatic burden of the disease and potential adverse effects. In our patients, the median interval of 17 treatment cycles was 5.0 [IQR 5.0, 7.5] weeks. In the ADAPT study, the median treatment cycle interval was 7.3 weeks [11,12], whereas in the ADAPT+ study, the average interval between cycles was 40.4 days, equivalent to 5.8 weeks [13,14]. In our patients, the treatment cycle interval might be short compared with that in the ADAPT and ADAPT+, because ADAPT/ADAPT+ subsequent cycle restarts in previous studies had to be conducted once the improved MG-ADL scale returned to near baseline level and treatment cycle intervals had to be maintained over 4 or 5 weeks.

The correlation between antibody levels and symptom scores in MG treatment suggests their potential as reference indexes for setting treatment cycle intervals. As treatment with efgartigimod progresses, continuous monitoring of the patients’ QMG score and IgG and AChR-Ab levels is essential. Improvements in these parameters may indicate the need to adjust the treatment cycle interval, aiming to maintain or improve the patients’ QOL. Conversely, an increase in the QMG score or antibody levels may necessitate more frequent administration of efgartigimod. For instance, in Case 1, reducing the treatment interval led to a gradual decrease in antibody titers. Meanwhile, Case 3 demonstrated that even a symptom fluctuation due to an interruption in treatment could be effectively managed by resuming the treatment cycle without additional medical interventions.

The importance of a patient-centered approach is exemplified in Case 2. The patient showed a modest improvement in the QMG score. However, substantial relief from long-term diplopia and external ophthalmoplegia highlights the necessity of considering not just clinical measurements but also symptomatic relief affecting the patients’ QOL, such as extraocular muscle palsy [25]. This finding aligns with that reported previously, suggesting that efgartigimod treatment notably improves ocular symptoms in patients with MG [26]. Similarly, Case 4 underscores the drug’s potential in managing bulbar symptoms. The patient experienced stabilized dysphagia symptoms despite initial treatment failures with other modalities such as plasmapheresis and intravenous methylprednisolone. This outcome suggests that efgartigimod has the potential efficacy, possibly even surpassing that of combination therapies that act immediately, for example, plasmapheresis, IVIg, and high-dose corticosteroid therapy like the case described in references [27]. Comparing the response of Japanese AChR-Ab-positive and SNMG patients to treatment with efgartigimod, it has been reported that AChR-Ab-positive patients have a higher response rate in subsequent cycles and a more sustained response to treatment. In contrast, responses in seronegative patients were less consistent and less durable [28]. The mild response to treatment with efgartigimod in Case 2 was consistent with this feature.

Moreover, recently, fatigue in generalized myasthenia gravis (MG) has been noticed as a symptom with a significant impact on quality of life (QoL). Zilucoplan, a cyclic peptide that binds to the protein complement, demonstrated statistically and clinically significant improvements in fatigue scores using Neuro-QoL Short Form Fatigue [29] and severity in the RAISE and RAISE-XT studies. These results suggest that Zilucoplan has a long-term ameliorating effect on fatigue in MG patients [30]. However, C5 inhibitors are available only for AChR-Ab-positive patients. On the other hand, efgartigimod is available for MuSK-Ab-positive patients and is expected to improve the severity of symptoms [28] and lead to recovery from fatigue in these patients.

Individualization of the efgartigimod administration schedule is imperative, taking into account each patient’s unique response to treatment, tolerance, and overall clinical stability. Regular follow-up and testing are crucial for making informed decisions about treatment cycle intervals, aiming to achieve an optimal balance between efficacy and safety. Additionally, considering the burden of frequent hospital visits on patients with MG, who often experience fatiguability, the development of efgartigimod for self-administration, particularly in subcutaneous formulations, could offer substantial benefits. Ongoing clinical trials, such as NCT04735432 and NCT04818671, may provide further insights into this aspect.

Recently, beyond MG, the therapeutic effect of efgartigimod on Stiff-Person Syndrome (SPS) has been reported [31]. SPS is an autoimmune neurological disease characterized by severe muscle rigidity and spasticity. Since the pathophysiology of SPS involves autoantibodies similar to those of MG, FcRn-targeted therapies, such as efgartigimod, can modify the disease process by reducing pathogenic IgG levels. This approach offers a new therapeutic strategy for SPS patients with limited treatment options and suggests a broad therapeutic range for efgartigimod.

This study has several limitations. First, this study is the age distribution of the patients. Three of the four patients included in this case series were relatively old compared to the average onset of myasthenia gravis. This demographic characteristic may have influenced the results due to associated comorbidities. In general, aging is correlated with a site-specific loss of skeletal muscle mass in both the extremities and trunk, and age-related loss of skeletal muscle mass and its function is also known as sarcopenia. Recent evidence has highlighted the impact of comorbidities on the management and outcome of MG [32]. Generalized EOMG in this study showed a more pronounced response to treatment with efgartigimod. The g-LOMGs also showed the expected benefit, but symptoms may include age-related muscle weakness, which might necessitate a higher treatment intensity when aiming to improve severity to minimal manifestations as a treatment goal. High-intensity treatment intensity may result in more severe low IgG and require strict monitoring concerning infection outbreaks. Potential variability of treatment efficacy in patients in different age group populations should be validated to promote the optimization of individualized treatment strategies for myasthenia gravis.

Second, efgartigimod treatment is used for MuSK-Ab-positive patients with MG in Japan; we did not assess treatment intervals in these patients, limiting insights into optimal management for this subgroup. Further research is needed to address these gaps and improve treatment strategies for all types of MG.

In conclusion, the real-world application of efgartigimod in MG treatment underscores the advantages of an individualized treatment cycle strategy. Adjusting treatment cycle intervals based on clinical response and disease progression allows for a more personalized approach to patient care, offering more substantial benefits than the scheduled additional administration if the MG-ADL scale does not improve to near baseline levels in the trials. However, continued research is essential to understand and optimize this treatment approach fully.

## Figures and Tables

**Figure 1 biomedicines-12-01214-f001:**
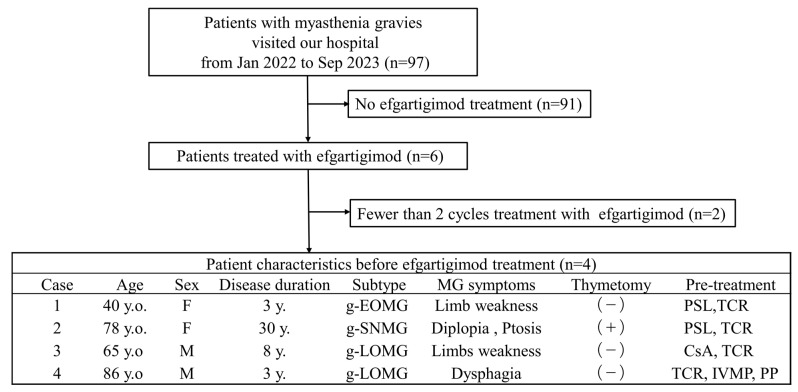
Flowchart showing the selection process for patients with myasthenia gravis included in this study on the efficacy of efgartigimod treatment. QMG score, quantitative myasthenia gravis score; y.o., years old; y., year; g-EOMG, generalized early-onset myasthenia gravis; g-LOMG, generalized late-onset myasthenia gravis; g-SNMG, generalized seronegative myasthenia gravis; PSL, prednisolone; TCR, tacrolimus; CsA, cyclosporine; IVMP, intravenous methylprednisolone; PP, plasmapheresis.

**Figure 2 biomedicines-12-01214-f002:**
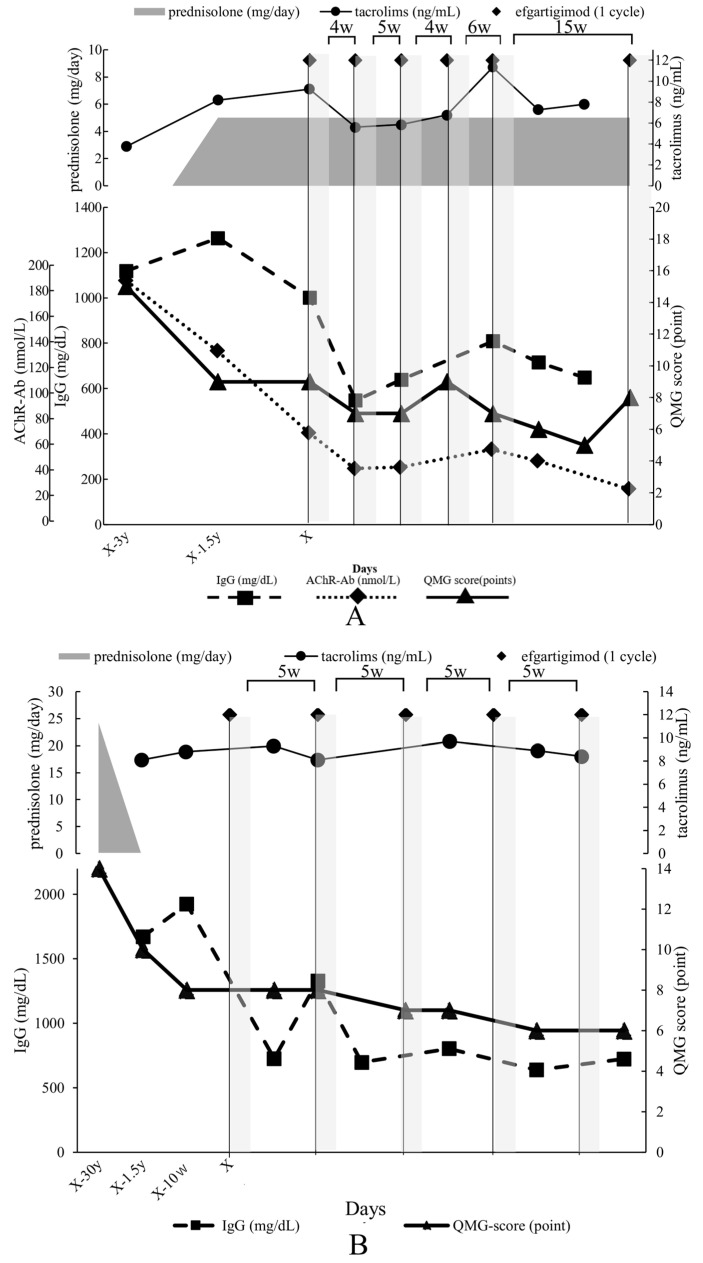
Graphs showing the clinical time course of patients with myasthenia gravis: (**A**) Case 1, (**B**) Case 2, (**C**) Case 3, and (**D**) Case 4. The horizontal axis represents the days before and after the start of efgartigimod treatment (y, year). “X” means the first treatment cycle of efgartigimod. The left vertical axis shows the IgG level in mg/dL (dotted line with square markers) and acetylcholine receptor antibody level in nmol/L (dotted line with diamond markers), and the right vertical axis shows the quantitative myasthenia gravis score (solid line with triangle markers). The gray-shaded area indicates the duration of the efgartigimod treatment cycle. Prednisolone dose (mg/day) is shown as a gray line, and tacrolimus concentration (ng/mL) is shown as a circled black line. The periods depicted above the figure indicate cycle intervals of efgartigimod (i.e., the period from the last infusion).

## Data Availability

The data are not publicly available as the information contained could compromise the privacy of the patients. The data that support the findings of this study are available, except for patients’ personal information, on request from the corresponding author, Shingo Konno.

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
