# Peer review of "Real-World Case Series of Efgartigimod for Japanese Generalized Myasthenia Gravis: Well-Tailored Treatment Cycle Intervals Contribute to Sustained Symptom Control"

_biomedicines, 2024, doi:10.3390/biomedicines12061214_

Round 1

Reviewer 1 Report

Comments and Suggestions for Authors

This is a nice case report on 4 treated MG patients with efgartigimod. Since this medication is not in use for antibody-negative MG in several countries, these informations are of great value for the neurmuscular community.

THe followiong minor problems shoul be addressed:

-in the discussion on page 7, as from line 200 and 206, nearly the same sentence "Determining a safe range..."

-Fig 2D: the intervals are not exactly beginning after last efgartigimod administration

-page 7, line 211: citation [17] is wrong, must be [18]

-please discuss the HP gastritis as an infection instead of saying none of your patients had any infections

I would recommend to emphasize/discuss differences between antibody positive and antibody negative cases as far as possible as well as the need for finding a medication that eases fatigue for the patients in the light of their cases

Comments on the Quality of English Language

Fig 2C "cyclispoline" used twice in the figure

line 214: cellar immunity...probably cellular immunity

line 225 grammar in "...must consider"

Author Response

Response to Reviewers' Comments

Manuscript ID: biomedicines-3013546
Title: Real-World Case Series of Efgartigimod for Generalized Myasthenia Gravis: Well-Tailored Treatment Cycle Intervals Contribute to Sustained Symptom Control

We greatly appreciate the insightful comments and suggestions provided by the reviewers. Following your feedback, we have implemented several revisions to enhance the clarity and accuracy of our manuscript. Below, we detail our responses to each point raised, indicating the changes made to the manuscript accordingly.

Reviewer #1

Major Comments:

  1. -in the discussion on page 7, as from line 200 and 206, nearly the same sentence "Determining a safe range..."
    The sentence "Determining a safe range..." was indeed repetitive. We have removed the duplicate instance to improve readability.

Revised in: Manuscript, Page 8, Line 248

  1. Fig 2D: the intervals are not exactly beginning after last efgartigimod administration

The intervals have been adjusted to commence immediately following the last administration of efgartigimod, as suggested. Please contact us again if there is a discrepancy between your intended modification and the revised figure.

Revised in: Figure 2D

  1. page 7, line 211: citation [17] is wrong, must be [18]

The citation has been corrected from [17] to [19] to reflect the accurate source.

Revised in: Manuscript, Page 9, Line 227

  1. please discuss the HP gastritis as an infection instead of saying none of your patients had any infections

We have elaborated on the relationship between low IgG levels and H. pylori infection, incorporating evidence that highlights the role of IgG and IgA antibodies in the immune response to H. pylori.

Revised in: Manuscript P.9, line 254-260.

1) Patients with hypogammaglobulinemia (low levels of immunoglobulins including IgG) show decreased serum bactericidal activity against H. pylori compared to healthy individuals [1]. This indicates that antibodies, especially IgG, play a role in the immune response against H. pylori.

2) While IgG antibodies against H. pylori are useful diagnostic markers, some studies found that IgA antibodies may be more strongly associated with an increased risk of gastric cancer in H. pylori infected patients [2]. This suggests IgA may be more relevant than IgG in the pathogenesis.

3) The search results do not directly state that low IgG levels lead to more severe or aggravated H. pylori infections [1]. The impaired bactericidal activity in hypogammaglobulinemia patients suggests low antibody levels could allow easier persistence of the infection.

In summary, while IgG antibodies contribute to the immune response against H. pylori, low IgG levels alone do not appear to definitively aggravate the infection based on these studies. The role of other antibody isotypes like IgA seems important as well.

[1] Desar IM, van Deuren M, Sprong T, Jansen JB, Namavar F, Vandenbroucke-Grauls CM, et al. Serum bactericidal activity against Helicobacter pylori in patients with hypogammaglobulinaemia. Clin Exp Immunol. (2009) 156:434-9.

[2] Aromaa A, Kosunen TU, Knekt P, Maatela J, Teppo L, Heinonen OP, Härkönen M, Hakama MK. Circulating anti-Helicobacter pylori immunoglobulin A antibodies and low serum pepsinogen I level are associated with increased risk of gastric cancer. Am J Epidemiol. (1996) 144:142-9.

  1. I would recommend to emphasize/discuss differences between antibody positive and antibody negative cases as far as possible

Additional text has been included in both the introduction and discussion sections, delineating the characteristics of MG subtypes and their response to treatment.

Revised in: Manuscript, Page 1, Line 36 – Page 2, Line 60 and Page 9, Line 299 - Page 10, Line 304

“The diversity of MG is further delineated by its classification into subtypes based on onset age, antibody status, and thymic pathology, each presenting unique management challenges. In terms of the subtypes we present, generalized Early-Onset MG (g-EOMG) typically manifests before age 50 and is often associated with thymic hyperplasia, with patients commonly having antibodies against the acetylcholine receptor. In contrast, Generalized Late-Onset MG (g-LOMG) appears after the age of 50 and often lacks thymic abnormalities, potentially presenting more severe symptoms and differing responses to treatments compared to earlier-onset forms. Another subtype, Generalized Seronegative MG (g-SNMG), is marked by the absence of typical AChR and MuSK antibodies, complicating both diagnosis and treatment due to the lack of specific serological markers [1]”.

[1] Akaishi T, Suzuki Y, Imai T, Tsuda E, Minami N, Nagane Y, et al. Response to treatment of myasthenia gravis according to clinical subtype. BMC Neurol. 2016 Nov 17;16(1):225.

“Comparing the response of Japanese AChR-Ab-positive and SNMG patients to treatment with efgartigimod, it has been reported that AChR-Ab-positive patients have a higher response rate in subsequent cycles and a more sustained response to treatment. In contrast, responses in seronegative patients were less consistent and less durable [1]. The mild response to treatment with efgartigimod in case 2 was consistent with this feature.”

[1] Suzuki S, Uzawa A, Nagane Y, Masuda M, Konno S, Kubota T, et al, Therapeutic Responses to Efgartigimod for Generalized Myasthenia Gravis in Japan. Neurol Clin Pract. (2024) 14: e200276.

  1. I would recommend to emphasize/discuss as well as the need for finding a medication that eases fatigue for the patients in the light of their cases.

Thank you for highlighting the importance of addressing fatigue in generalized MG. We acknowledge that fatigue significantly impacts the quality of life in these patients.

Revised in: Manuscript, Page 10, Line 305-313

“Moreover, recently, fatigue in generalized myasthenia gravis (MG) has been noticed as a symptom with a significant impact on quality of life (QoL). Zircoplan, a cyclic peptide that binds to the protein complement, demonstrated statistically and clinically significant improvements in fatigue scores using Neuro-QoL Short Form Fatigue [1] and severity in the RAISE and RAISE-XT studies. These results suggest that Zircoplan has a long-term ameliorating effect on fatigue in MG patients [2]. However, C5 inhibitors are available only for AChR-Ab positive patients. On the other hand, efgartigimod is available for MuSK-Ab positive patients and is expected to improve the severity of symptoms [3] and lead to recovery from fatigue in these patients.”

[1] Cella D, Lai JS, Nowinski CJ, Victorson D, Peterman A, Miller D, et al. Neuro-QOL: brief measures of health-related quality of life for clinical research in neurology. Neurology. (2012) 78:1860-7.

[2] Weiss MD, Freimer M, Leite MI, Maniaol A, Utsugisawa K, Bloemers J, et al. Improvement of fatigue in generalised myasthenia gravis with zilucoplan. J Neurol. (2024) 271:2758-67.

[3]Suzuki S, Uzawa A, Nagane Y, Masuda M, Konno S, Kubota T, et al, Therapeutic Responses to Efgartigimod for Generalized Myasthenia Gravis in Japan. Neurol Clin Pract. (2024) 14: e200276.

Mainor Comments:

  1. Language Corrections:

We have made the following corrections to address the language concerns:

"cyclispoline" to "cyclosporine"

"cellar immunity" to "cellular immunity"

"must consider" to "must be considered"

Revised in: Manuscript, Figure2D, Page 9, Line 250 and Page 9, Line 267

Reviewer 2 Report

Comments and Suggestions for Authors

In the introduction, insert description of the subtypes: g-EOMG, g-SNMG and g-LOMG. The individual figures of the 4 cases are interesting, but interesting is to create a figure with efgartigimod treatment of all 4 cases plus a median value of the literature data to conrfornte.

Author Response

Response to Reviewers' Comments

Manuscript ID: biomedicines-3013546

Title: Real-World Case Series of Efgartigimod for Generalized Myasthenia Gravis: Well-Tailored Treatment Cycle Intervals Contribute to Sustained Symptom Control

We greatly appreciate the insightful comments and suggestions provided by the reviewers. Following your feedback, we have implemented several revisions to enhance the clarity and accuracy of our manuscript. Below, we detail our responses to each point raised, indicating the changes made to the manuscript accordingly.

Reviewer #2

Major Comments:

1.1n the introduction, insert description of the subtypes: g-EOMG, g-SNMG and g-LOMG.

We have added descriptions of MG subtypes as requested to better contextualize their unique management challenges and implications for treatment.

Revised in: Manuscript, Page 1, Line 36 – Page 2, Line 56

“The diversity of MG is further delineated by its classification into subtypes based on onset age, antibody status, and thymic pathology, each presenting unique management challenges. In terms of the subtypes we present, generalized Early-Onset MG (g-EOMG) typically manifests before age 50 and is often associated with thymic hyperplasia, with patients commonly having antibodies against the acetylcholine receptor. In contrast, Generalized Late-Onset MG (g-LOMG) appears after the age of 50 and often lacks thymic abnormalities, potentially presenting more severe symptoms and differing responses to treatments compared to earlier-onset forms. Another subtype, Generalized Seronegative MG (g-SNMG), is marked by the absence of typical AChR and MuSK antibodies, complicating both diagnosis and treatment due to the lack of specific serological markers [1]”.

[1] Akaishi T, Suzuki Y, Imai T, Tsuda E, Minami N, Nagane Y, et al. Response to treatment of myasthenia gravis according to clinical subtype. BMC Neurol. 2016 Nov 17;16(1):225.

2.The individual figures of the 4 cases are interesting, but interesting is to create a figure with efgartigimod treatment of all 4 cases plus a median value of the literature data to conrfornte.

Thank you for your fascinating points. We reviewed the results of the representative ADPT trials and found that none of the enrolled patients' treatment cycle intervals and changes in QMG scores were shown in one figure, as in our figure. Nor were the changes in AChR-Ab antibodies noted. Therefore, it was challenging to create a figure that combined the four cases and the data in the literature, as you suggested.

Reviewer 3 Report

Comments and Suggestions for Authors

This interesting article gives a very interesting perspective on the use of efgartigimod in MG. The article is well-written and punctual. I think that this paper might be very useful, but a revision of some points is necessary:

·      Lines 64-65. As the indication is also for seronegative MG, it must be indicated in the title that the real-life comes from Japan.

·      Results. The four patients presented are mostly quite old considering the mean age of the disease, hence results may be influenced by relevant comorbidities. Indeed, it is reasonable to expect a prominent response from young patients with high antibodies titres rather than from older patients with low titres and low autoimmunity burden. I suggest onsidering it in the limitations and discuss on the light of recent evidences (Comorbidity in myasthenia gravis: multicentric, hospital-based, and controlled study of 178 Italian patients. Neurol Sci. 2024).

·      The discussion should be widened. I suggest discussing the advantages of efgartigimod when compared with others treatments:

o   IVIg.The problem with IVIg is not only the cost for production, but also the lack of volunteer blood donors as IVIg is a product from elaboration of plasma. Hence, there is low availability, also because IVIg are used for many other indications (CIDP, hematology, etc).

o   PLEX. Efgartigimod acts similarly to PLEX reducing total IgG (and pathogenetic antibodies) serum levels. As the mechanism is very similar to PLEX, efgartigimod acts very fast and early amelioration is achieved when compared to IVIg or other immunotherapies. The problem with venous accesses in PLEX should be discussed.

o   B-depletive therapies and severe immunosuppression

o   Complement inhibitors and need for vaccinations

·      I suggest discussing the potential of efgartigimod in autoimmune comorbidity in patients affected by MG. Of note, there very recent data on improved autoimmune comorbidity by efgartigimod treatment, a recent study with brilliant results on MG with comorbid stiff-person syndrome: Discuss its role citing recent evidence (Efgartigimod beyond myasthenia gravis: the role of FcRn-targeting therapies in stiff-person syndrome. J Neurol. 2023).

·      The median administration interval of 5 weeks is a quite aggressive schedule. However, it was demonstrated that EFG does not allow reduction over 60%. Hence 40% of IgG from the starting dose should always be > 100 mg. What do the authors think about this? How many patients dropped to 100 mg of IgG? Did they experience any infection?

·      The article presents some grammar mistakes. I suggest revising them.

o   Line 224. Be considered?

o   Line 254. Remove one efgartigimod

Comments on the Quality of English Language

minor check required

Author Response

Response to Reviewers' Comments

Manuscript ID: biomedicines-3013546

Title: Real-World Case Series of Efgartigimod for Generalized Myasthenia Gravis: Well-Tailored Treatment Cycle Intervals Contribute to Sustained Symptom Control

We greatly appreciate the insightful comments and suggestions provided by the reviewers. Following your feedback, we have implemented several revisions to enhance the clarity and accuracy of our manuscript. Below, we detail our responses to each point raised, indicating the changes made to the manuscript accordingly.

Reviewer #3

Major Comments:

  1. Lines 64-65. As the indication is also for seronegative MG, it must be indicated in the title that the real-life comes from Japan.

The word "Japanese" has been inserted into the title to specify the geographic focus of the real-world data presented.

Revised in: Manuscript, Title

  1. Results. The four patients presented are mostly quite old considering the mean age of the disease, hence results may be influenced by relevant comorbidities. Indeed, it is reasonable to expect a prominent response from young patients with high antibodies titters rather than from older patients with low titters and low autoimmunity burden. I suggest considering it in the limitations and discuss on the light of recent evidences (Comorbidity in myasthenia gravis: multicentric, hospital-based, and controlled study of 178 Italian patients. Neurol Sci. 2024).

We have acknowledged the potential influence of age and comorbidities on treatment response in our limitations section, citing recent studies that discuss these factors.

Revised in: Manuscript, Page 10, Line 330 - 344

“This study is the age distribution of the patients. Three of the four patients included in this case series were relatively old compared to the average onset of myasthenia gravis. This demographic characteristic may have influenced the results due to associated comorbidities. In general, aging is correlated with a site-specific loss of skeletal muscle mass in both the extremities and trunk, and age-related loss of skeletal muscle mass and its function is also known as sarcopenia. Recent evidence has highlighted the impact of comorbidities on the management and outcome of MG [1] Generalized-EOMG in this study showed a more pronounced response to treatment with efgartigimod. The g-LOMGs also showed the expected benefit, but symptoms may include age-related muscle weakness, which might necessitate a higher treatment intensity when aiming to improve severity to minimal manifestations as a treatment goal. High-intensity treatment intensity may result in more severe low IgG and require strict monitoring concerning infection outbreaks. Potential variability of treatment efficacy in patients in different age group populations should be validated to promote the optimization of individualized treatment strategies for myasthenia gravis.”

[1] Di Stefano V, Iacono S, Militello M, Leone O, Rispoli MG, Ferri L, et al, Comorbidity in myasthenia gravis: multi-centric, hospital-based, and controlled study of 178 Italian patients. Neurol Sci. 2024 Feb 22. doi: 10.1007/s10072-024-07368-0

  1. The discussion should be widened. I suggest discussing the advantages of efgartigimod when compared with others treatments

We have expanded our discussion to better highlight the distinct advantages of efgartigimod over other treatments like intravenous gamma globulin (IVIg), plasmapheresis, B-cell depletive therapies, and complement inhibitors

Revised in: Manuscript, Page 8, Lines 216-232

 “Efgartigimod distinguishes itself from other treatment options such as intravenous gamma globulin (IVIg), plasmapheresis, B-cell depletive therapies, and complement inhibitors due to its unique mechanism of action and clinical advantages. IVIg, despite its effectiveness, faces challenges like high production costs and dependence on

volunteer blood donors for plasma extraction, leading to limited availability for treating various medical conditions like chronic inflammatory demyelinating polyneuropathy and hematological disorders. In contrast, Efgartigimod rapidly reduces total IgG and pathogenic antibody levels in the serum, offering quicker and more immediate

improvement compared to IVIg and other immunotherapies. Similarly to plasmapheresis, Efgartigimod lowers IgG levels without the logistical issues associated with venous access, providing a smoother treatment experience. Unlike B-cell depletive therapies and severe immunosuppression, which carry the risk of increased infections and com-plications due to extensive immunosuppression, Efgartigimod achieves IgG reduction without inducing such profound immunosuppression, potentially offering a safer treatment option. Additionally, while complement inhibitors are effective, they require extensive vaccination protocols due to their specific mechanism of action, whereas

Efgartigimod does not necessitate such vaccinations, making it a more convenient choice for patients and healthcare providers.”

  1. I suggest discussing the potential of efgartigimod in autoimmune comorbidity in patients affected by MG. Of note, there very recent data on improved autoimmune comorbidity by efgartigimod treatment, a recent study with brilliant results on MG with comorbid stiff-person syndrome: Discuss its role citing recent evidence (Efgartigimod beyond myasthenia gravis: the role of FcRn-targeting therapies in stiff-person syndrome. J Neurol. 2023).

We have included recent findings on efgartigimod's broader therapeutic potential, particularly its promising effects in treating Stiff-Person Syndrome (SPS), an autoimmune condition that shares some pathophysiological features with myasthenia gravis. This addition underscores efgartigimod’s potential as a versatile therapy in the landscape of autoimmune diseases, beyond its primary application in MG.

Revised in: Manuscript, Page 10, Lines 323-329

“Recently, beyond MG, the therapeutic effect of efgartigimod on Stiff-Person Syndrome (SPS) has been reported [26]. SPS is an autoimmune neurological disease characterized by severe muscle rigidity and spasticity. Since the pathophysiology of SPS involves autoantibodies similar to those of MG, FcRn-targeted therapies, such as efgartigimod, can modify the disease process by reducing pathogenic IgG levels. This approach offers a new therapeutic strategy for SPS patients with limited treatment

options and suggests a broad therapeutic range for efgartigimod.”

5.The median administration interval of 5 weeks is a quite aggressive schedule. However, it was demonstrated that EFG does not allow reduction over 60%. Hence 40% of IgG from the starting dose should always be > 100 mg. What do the authors think about this? How many patients dropped to 100 mg of IgG? Did they experience any infection?

All patients presented with subnormal hypogammaglobulinemia, but at least >100 mg/dL was maintained throughout the course of treatment.

Revised manuscript: P.9, line 251.

In case 3, a gastric ulcer appeared during treatment with efgartigimod and H. pylori infection was diagnosed. However, we do not believe that low IgG associated with treatment with efgartigimod is necessarily the cause of gastric ulceration (revised manuscript P.9, line 254-260). The reasons are as follows;

1) Patients with hypogammaglobulinemia (low levels of immunoglobulins including IgG) show decreased serum bactericidal activity against H. pylori compared to healthy individuals [1]. This indicates that antibodies, especially IgG, play a role in the immune response against H. pylori.

2) While IgG antibodies against H. pylori are useful diagnostic markers, some studies found that IgA antibodies may be more strongly associated with an increased risk of gastric cancer in H. pylori infected patients [2]. This suggests IgA may be more relevant than IgG in the pathogenesis.

3) The search results do not directly state that low IgG levels lead to more severe or aggravated H. pylori infections [1]. The impaired bactericidal activity in hypogammaglobulinemia patients suggests low antibody levels could allow easier persistence of the infection.

In summary, while IgG antibodies contribute to the immune response against H. pylori, low IgG levels alone do not appear to definitively aggravate the infection based on these studies. The role of other antibody isotypes like IgA seems important as well.

[1] Desar IM, van Deuren M, Sprong T, Jansen JB, Namavar F, Vandenbroucke-Grauls CM, van der Meer JW. Serum bactericidal activity against Helicobacter pylori in patients with hypogammaglobulinaemia. Clin Exp Immunol. 2009 Jun;156(3):434-9.

[2] Aromaa A, Kosunen TU, Knekt P, Maatela J, Teppo L, Heinonen OP, Härkönen M, Hakama MK. Circulating anti-Helicobacter pylori immunoglobulin A antibodies and low serum pepsinogen I level are associated with increased risk of gastric cancer. Am J Epidemiol. 1996 Jul 15;144(2):142-9.

Mainor Comments:

  1. The article presents some grammar mistakes. I suggest revising them.

Line 224. Be considered? Line 254. Remove one efgartigimod

To ensure the manuscript meets the highest standards of academic writing, we have corrected noted grammatical errors and improved the language for better readability and professional tone. Specific corrections include altering "must consider" to "must be considered" and removing an extraneous mention of "efgartigimod."

Revised manuscript; Page 8, Lines 242 and 296
